# Survival of *Lactobacillus rhamnosus* GG in Chitosan-Coated Alginate Beads: Effects of Food Matrices (Casein, Corn Starch, and Soybean Oil) and Dynamic Gastrointestinal Conditions [note 1]

**DOI:** 10.3390/foods14122094

**Published:** 2025-06-13

**Authors:** Toshifumi Udo, Gopinath Mummaleti, Zijin Qin, Jinru Chen, Rakesh K. Singh, Yang Jiao, Fanbin Kong

**Affiliations:** 1Department of Food Science and Technology, University of Georgia, Athens, GA 30602, USA; toshifumi.udo@uga.edu (T.U.); gopinath.mummaleti@uga.edu (G.M.); zijin.qin@uga.edu (Z.Q.); jchen@uga.edu (J.C.); rsingh@uga.edu (R.K.S.); 2College of Food Science and Technology, Shanghai Ocean University, Shanghai 201306, China; yjiao@shou.edu.cn

**Keywords:** Probiotics, *Lactobacillus rhamnosus* GG, alginate, chitosan coating, dynamic gastrointestinal models, extrusion, cross-linking, casein, corn starch, soybean oil

## Abstract

Probiotics like *Lactobacillus rhamnosus* GG (LRGG) offer health benefits but face reduced viability under harsh gastrointestinal (GI) conditions. Encapsulation improves stability, yet most studies rely on static GI models with a simplified environment that may overestimate survival. This study assessed LRGG survival using chitosan-coated alginate beads under both static and dynamic GI models, including peristaltic flow and continuous juice replenishment. Food matrices (casein, corn starch, and soybean oil) were tested in static models. Beads were prepared via extrusion and subjected to simulated gastric and intestinal digestion. After 2 and 4 h of digestion, casein preserved LRGG viability at 8.50 ± 0.11 Log CFU/g, compared to 5.81 ± 0.44 with starch and undetectable levels with soybean oil. Casein’s protective effect was attributed to its pH-buffering capacity, raising gastric pH from 2.5 to 4.6. Starch offered moderate protection, while soybean oil led to bead dissolution due to destabilization of the egg-box structure. Dynamic GI models showed greater reductions in LRGG viability than static models, emphasizing the need for physiologically relevant simulations. The results highlight the importance of selecting appropriate food matrices and digestion models for accurate probiotic assessment, supporting improved encapsulation strategies in functional food development.

## 1. Introduction

Probiotics are live microorganisms that confer health benefits to the host when consumed in adequate amounts [1,2,3]. They play a crucial role in maintaining gut microbiota balance, enhancing immune function, and preventing gastrointestinal (GI) disorders [4,5,6]. Among them, *Lactobacillus rhamnosus* GG (LRGG) is one of the most well-documented probiotic strains, widely recognized for its ability to improve gut health and provide resistance against pathogenic infections [7,8,9]. However, to exert their beneficial effects, probiotics must survive the harsh conditions of the GI tracts, including exposure to gastric acid, digestive enzymes, and bile salts [10,11,12,13]. A significant challenge in probiotic applications is ensuring their viability throughout digestion and storage, which has led to increasing interest in encapsulation technologies as a protective strategy.

Encapsulation involves enclosing probiotics within protective materials to shield them from adverse external factors, such as acidity, oxidation, and mechanical stress [14,15,16]. From a commercial perspective, encapsulated probiotics have vast potential across functional foods, nutraceuticals, and pharmaceuticals. The global probiotic market was valued at over USD 87.7 billion in 2023 and is projected to exceed USD 100 billion by 2030, with encapsulated probiotics comprising a growing share due to demand for targeted delivery systems and shelf-stable products [17,18]. There is a significant market demand for encapsulation solutions that extend probiotic viability during processing, storage, and digestion, especially in non-dairy, plant-based, or shelf-stable formulations. Various encapsulation techniques have been explored, including spray drying, extrusion, emulsion-based methods, and hydrogel entrapment, each offering different levels of protection and controlled release [19,20]. A trend of encapsulation with nanomaterials is another emerging area of study, particularly for enhancing the antimicrobial and functional properties of packaging and delivery systems. For instance, bionanocomposites such as silver nanoparticles have been investigated for active food packaging applications due to their shelf-life-extending and protective functions, which may inform future directions in probiotic delivery as well [21]. Alginate-based encapsulation is among the most widely studied approaches due to its biocompatibility, cost-effectiveness, and ability to form gel matrices that enhance probiotic stability with versatile, extensive approaches [22,23,24]. Furthermore, chitosan-coated alginate beads provide additional protection, as chitosan enhances resistance to acidic environments and bile salts, improving probiotic survival during digestion [22,25,26,27].

In addition to encapsulation, the presence of food materials in the digestive environment may significantly impact probiotic survival and release. Specific food components, such as proteins, carbohydrates, and lipids, may provide additional protection to probiotics by altering the local digestive environment and interacting with the encapsulation matrix. For instance, protein is known to exhibit pH-buffering capacities, potentially creating a situation that mitigates gastric acidity and then improves probiotic viability [28,29,30]. Starch is also a common material to improve the integrity of probiotic capsules as a filler or barrier matrix, limiting diffusion of gastrointestinal juices into the encapsulants [31,32]. Lipids are often incorporated into emulsion-based encapsulation methods and freeze-drying encapsulation technology, providing a carrier for hydrophobic compounds and carbon sources for probiotic fermentation [30,33]. However, their effects when present outside the capsules and digested simultaneously have not been thoroughly investigated. Since probiotics are typically consumed with food, simulating digestion in the presence of food materials offers a more relevant assessment of probiotic viability and supports better design of effective probiotic administration. Therefore, evaluating the role of basic food matrices in probiotic protection other than wall materials is also essential.

Moreover, static in vitro GI models have been traditionally used to evaluate the survival of encapsulated probiotics. These models provide controlled conditions, where pH, enzyme concentrations, and digestion times are maintained at predefined levels [14,34]. In contrast, dynamic in vitro GI models offer a more realistic simulation of digestion by incorporating gradual pH changes, continuous replenishment of digestive fluids, and peristaltic movements, mimicking the actual in vivo environment more accurately [35,36,37]. It has been revealed that the digestibility of in vitro dynamic GI models has more relevance than that of in vivo GI models [36,37]. There are also a few studies using dynamic GI models for assessing probiotics strains [38,39]; however, assessments of encapsulated probiotics were rarely conducted. While the use of dynamic GI models to evaluate food materials is gaining a huge interest in the area of food science, the impacts of dynamic physiological activities of GI tracts on encapsulated probiotics are still unclear.

Hereby, this study, evaluating encapsulated probiotics under more advanced digestive conditions, has two main objectives. One of the main aims of this study is to evaluate the effects of food matrices, such as casein, corn starch, and soybean oil, on the survivability and release of probiotics encapsulated in alginate-based capsules with chitosan coating in GI conditions. Second, this study aims to assess chitosan-coated alginate probiotic capsules using in vitro dynamic GI models simulating continuous GI juice supply and emptying, which replicate one of the key physiological activities in the human GI tracts. The findings from this research will provide valuable insights into the optimization of encapsulation techniques, the role of food matrices in probiotic protection, and the importance of dynamic GI models in assessing probiotic stability. This study contributes to the development of more effective probiotic formulations for functional food applications, ensuring enhanced survival and bioavailability in the human digestive tract.

## 2. Materials

Freeze-dried *Lactobacillus rhamnosus* GG (LRGG) ATCC 53103 was obtained from commercial probiotic capsules (i-Health Inc., Shelton, CT, Denmark). Peptone was purchased from Becton, Dickinson and Company (Franklin Lakes, NJ, USA). MRS broth and MRS agar were obtained from Alpha Biosciences (Baltimore, MD, USA). Sodium alginate (SA) was purchased from bioWORLD (Dublin, OH, USA). Chitosan (50–100 mPa·s 0.5% in 0.5% acetic acid at 20 °C) was obtained from Tokyo Chemical Industry Co., Ltd. (Tokyo, Japan). Soybean oil and corn starch were obtained from grocery stores. Digestive enzymes and other chemicals used in this study were obtained from Sigma Aldrich (St. Louis, MO, USA).

## 3. Methods

### 3.1. Cell Preparation

Freeze-dried LRGG was diluted into a sterile 0.1% (*w*/*v*) peptone solution and incubated for 18 h at 37 °C. The suspended cells were then spread onto MRS agar and incubated for 18 h at 37 °C to obtain sub-cultures. Thereafter, the cultured colony was taken from the agar media and inoculated into MRS broth and incubated for 22 h at 37 °C to obtain 9–10 Log CFU/mL of proliferated probiotics. The cultured cells were harvested by centrifugation at 4400× *g* and then washed twice with sterile 0.1% peptone solution. Finally, the collected cells were resuspended in a sterile 0.1% peptone solution to adjust the concentration to 10 Log CFU/mL.

### 3.2. Encapsulation of Lactobacillus rhamnosus GG by Ionic Gelation

The encapsulation of LRGG was performed using a previously published method with modifications [40]. The cell suspension was mixed with sterile 2.0% *w*/*v* alginate solution at a volume ratio of 1:4 before the extrusion. The mixture was then loaded into a 30 mL syringe with a 22 G needle, followed by dropwise extrusion into a 0.1 M sterile CaCl_2_ solution at a rate of 5 mL/min using a syringe pump. The beads were placed for 30 min for further external gelation. Some beads were subsequently immersed in a sterile 0.4% (*w*/*w*) low molecular chitosan solution for 40 min to obtain chitosan-coated alginate microcapsules. After the gelation and coating, the beads were sieved, washed, and stored in a 0.1% peptone solution at 4 °C until use. All beads were used for later tests within 2 days. Viable LRGG cells were enumerated using a standard plate count with serial dilution, followed by incubation on MRS agar. The encapsulation efficiency (EE) was calculated using the following equation.(1)$$EE\;\left(\%\right)=\frac{\log N}{\log N_{0}}\times 100$$ where *N* is the number of viable cells in beads (Log CFU/g) and *N*_0_ is the number of initial cells added into the samples (Log CFU/g).

### 3.3. Viability of Encapsulated Probiotics in In Vitro Static/Dynamic Gastrointestinal Digestion Models

The viability of the encapsulated LRGG was tested under both static and dynamic conditions. Simulated gastric fluid was prepared according to a previously published paper [41], where it contains 5 mg/mL pepsin (from porcine gastric mucosa, P7000-100G, Sigma Aldrich, St. Louis, MO, USA), 6 mg/mL mucin, 5.504 mg/mL NaCl, 1.648 mg/mL KCl, 0.532 mg/mL NaH_2_PO_4_, 0.798 mg/mL CaCl_2_·2H_2_O, 0.612 mg/mL NH_4_Cl, and 0.17 mg/mL urea. The stock solution was first prepared by filter sterilization with a 0.45 μm membrane filter. Pepsin and mucin were freshly added to the filtered solution right before the gastric testing. A total of 2 g of chitosan-coated alginate beads were mixed with 18 mL of the simulated gastric fluid in a 50 mL Erlenmeyer flask, and digestion tests were performed for 2 h by an in vitro static gastric digestion model (SGDM), where the mixture was incubated in a shaking bath at 37 °C and 150 rpm. A dynamic gastric digestion model (DGDM) was also tested, which simulated continuous secreting and emptying of gastric fluid in addition to the SGDMs (Figure 1). For the setup, a pair of peristaltic pumps was connected to sterilized tubes, and the tube ends were placed in the gastric model. The other ends of the tubes were placed in a gastric juice stock and a sterilized conical tube for juice supply and emptying. The emptying tube was closed with a filter paper to avoid emptying the beads from the model during the test. The supplying and emptying of gastric juice were continuously performed at a rate of 2 mL/min for 2 h. After the gastric tests, beads were sieved and washed with a sterile 0.1% peptone solution, and the final weight of the beads was measured. The weight ratio of beads was calculated by the equation shown below, indicating >100% as swelling and <100% as shrinking, where M_0_ is the initial weight and M is the weight after 2 h of gastric digestion.(2)$$Weight\;ratio\;\left(\%\right)=\frac{M}{M_{0}}\times 100$$

The weighed beads were immediately soaked in a 1.0% sterile citrate buffer (pH 6.5) and adjusted to a final volume of 10 mL. The beads were gently agitated on an orbital shaker until the beads were completely dissolved. The viability of probiotics was measured by a normal bacterial plate count using MRS agar plates for 48 h of incubation at 37 °C. The pH shifts in gastric juice in the SGDMs and DGDMs were also monitored every 30 min.

### 3.4. Release of Encapsulated Probiotics in Static/Dynamic Intestinal Digestion Models

After static gastric digestion for 2 h, the chitosan-coated alginate beads were transferred to another Erlenmeyer flask, and the gastric juice was replaced with the simulated intestinal fluid, which was prepared according to a previously published paper [42]. Briefly, the intestinal juice was prepared by mixing the simulated pancreatic juice (18 mg/mL porcine pancreatin, 3 mg/mL porcine pancreatic lipase, 14.024 mg/mL NaCl, 6.776 mg/mL NaHCO_3_, 1.128 mg/mL KCl, 160.0 mg/mL KH_2_PO_4_, 100.0 mg/mL MgCl_2_, and 0.2 mg/mL urea) and the simulated bile juice (60 mg/mL porcine bile extract, 10.518 mg/mL NaCl, 0.752 mg/mL KCl, 11.57 mg/mL NaHCO_3_, and 0.50 mg/mL urea) followed by adjusting the pH to 8.1 by 1.0 M HCl/NaOH. The release of probiotics from the alginate and alginate-chitosan capsules was tested using in vitro static intestinal digestion models (SIDMs) and in vitro dynamic intestinal digestion models (DIDMs). A total of 0.1 mL of intestinal fluid was collected every hour up to 4 h, and the viability of released probiotics was measured by a normal bacterial plate count using MRS agar plates for 48 h of incubation at 37 °C. The remaining viable cells in the chitosan-coated alginate beads were also measured as described above after 4 h of intestinal digestion.

### 3.5. Effect of Food Matrices on In Vitro Gastric Digestion Models

To investigate the effect of food matrices during gastric digestion, corn starch, soybean oil, and casein were additionally added for SGDMs and SIDMs. A total of 4 g of model food materials were mixed with 18 mL of simulated gastric fluid, followed by adding 2 g of alginate/chitosan-coated alginate beads. Gastric digestions were performed using SGDMs following the described methods above. Encapsulated probiotics without food matrices were also tested under the same conditions as controls. After 2 h of gastric digestion, food materials and chitosan-coated alginate beads were transferred to another Erlenmeyer flask and replenished with the simulated intestinal fluid.

### 3.6. Characterization of Probiotic Capsules

The appearance of microcapsules before and after in vitro gastric tests was observed by a digital microscope (ADSM302, Andonstar Technology Co., Ltd., Shenzhen, China). The diameter of initial beads was also calculated by Image J software (Version 1.53t, National Institutes of Health, Bethesda, MD, USA) from images of 20 randomly chosen microcapsules, as a circle equivalent. The average value was used as the diameter of the beads.

### 3.7. FTIR Spectroscopy

Fourier-transform infrared (FTIR) spectra were recorded using a Bruker Alpha II spectrometer operating in transmission mode. The samples were vacuum-dried for 24 h at 25 °C before measurements. Each sample was loaded on the stage, and measurements were carried out in the range of 4000 to 400 cm^−1^ with a spectral resolution of 4 cm^−1^. Each spectrum was obtained by averaging 35 scans for the sample and 64 scans for the background, with a total acquisition time of approximately 45 s per sample. All spectra were baseline-corrected and normalized prior to interpretation. Characteristic peaks were identified based on standard studies.

### 3.8. Statistical Analysis

All tests were performed independently in triplicate, and analysis of variance (ANOVA) was used for data analysis. One-way ANOVA with Tukey’s post hoc test and two-way repeated measures ANOVA with Bonferroni’s pairwise *t*-test were conducted using R (version 4.2.1). All the presented data are expressed as mean ± standard deviation, with a significance level at *p* < 0.05.

## 4. Results and Discussion

### 4.1. Impacts of Food Materials in Static Gastrointestinal Models

The alginate and chitosan-coated beads were prepared, and the EE of the beads was 98.70 ± 1.56 and 98.42 ± 0.65%, and the particle size was 2.32 ± 0.16 and 2.28 ± 0.13 mm, respectively (*p* > 0.05). The impact of food matrices on encapsulated probiotics was investigated using the static GI conditions. Figure 2a shows the amount of viable LRGG entrapped in alginate or chitosan-coated alginate beads after 2 h of digestion using SGDMs with model food materials, including soybean oil, corn starch, and casein, and samples without food materials as controls. Compared to the control alginate beads, the survivability of LRGG was significantly improved with chitosan-coated capsules, consistent with previous findings [25]. Casein increased LRGG viability from no viable cells detected to 8.65 ± 0.43 Log CFU/g in alginate beads and 6.34 ± 0.27 Log CFU/g to 8.79 ± 0.40 Log CFU/g in chitosan-coated beads (*p* < 0.05). This was assumed to be mainly due to its pH-buffering effect that mitigates acid stress during gastric digestion. Similar protection effects were observed when incorporated into a drug delivery system with casein [43] and other proteins, such as soy [44] and whey [45]. Other than casein, the viability of LRGG was significantly enhanced when the alginate beads were digested with corn starch to 7.87 ± 0.16, while cell counts dropped to the detection limit (<2 Log CFU/g) in both soybean oil and control samples.

The weight ratio, which was measured as swelling or shrinking rates, and pH shifts were also shown in Figure 2b and Figure 2c, respectively. For alginate beads, casein was the only food matrix that caused the beads to swell during gastric digestion. The weight ratios relative to the original weights were 89.3 ± 0.9% (control), 88.0 ± 1.3% (soybean oil), 73.2 ± 2.9% (starch), and 124.8 ± 1.3% (casein), with casein showing a significantly higher value (*p* < 0.05). Notably, the addition of casein had opposite effects on alginate and chitosan-coated alginate beads, where alginate beads swelled, while chitosan-coated beads shrank after 2 h of gastric digestion. The weight ratios of chitosan-coated beads were 92.0 ± 0.9% (control), 88.9 ± 3.8% (soybean oil), and 84.9 ± 3.8% (starch), with no significant differences among these groups (*p* > 0.05). However, the addition of casein led to a significant reduction in weight ratio to 47.2 ± 0.9% (*p* < 0.05), indicating pronounced shrinkage. This discrepancy may be attributed to the difference between the negatively charged alginate and the positively charged chitosan on their surfaces. It is known that the isoelectric point of casein is about 4.6 [46], which results in a positive charge on the protein during the gastric digestion. Regarding pH shifts, it was clearly indicated that the addition of casein significantly modulated the acidity of gastric juice compared to the control and other food materials (*p* < 0.05), increasing the pH from 2.5 to 4.6 after 2 h of gastric testing. Alginate beads showed a slightly lower pH compared to chitosan-coated beads after 2 h static gastric digestions (*p* < 0.05), while no significant difference was observed among the control, soybean oil, and starch in alginate beads (*p* > 0.05). The addition of corn starch slightly increased the pH in the gastric juice of chitosan-coated beads more than the control and soybean oil, while no significant improvement in survivability was shown. Considering the above results, it is supposed that, to improve the survival rate of probiotics, it would be more effective to simultaneously digest proteins such as casein due to strong pH buffering. Similar effects have been observed in the enhanced stability of probiotics in dairy products, for example, milk [47,48], yogurt [49], and cheese [50]. Therefore, this strategy could be widely applicable to various types of probiotic capsules. The addition of starch also improved the survival of alginate capsules, indicating the possibility of external protection of food materials other than pH modulation; however, it was believed that the protective effect is not as strong as that of proteins.

Figure 3 shows microscopic observations of the sample’s appearance before and after digestion. With the exception of the alginate beads with casein, bead shrinkage was observed in the other conditions, especially in the chitosan beads digested with casein, which corresponds to the weight ratio of each sample.

The FTIR spectra of alginate and chitosan-coated alginate beads, along with their corresponding digested samples after 2 h in SGDMs, are summarized in Figure 4a and Figure 4b, respectively. Both alginate and chitosan-coated beads exhibited some common peaks which are also found in studies, including a broad O–H stretching band around 3400–3500 cm^−1^ and a strong C=O band at 1640 cm^−1^, which are attributed to the hydroxyl and carboxyl groups present in the alginate structure, while peaks at 2932, 2862, and 1749 cm^−1^ were only detected in alginate beads [51,52,53]. In beads digested with casein, both types showed an additional band at 1547–1560 cm^−1^, likely corresponding to the Amide II N–H bending vibration of the casein protein [54]. A peak at 1007 cm^−1^ by C-O-C stretching was observed exclusively in alginate beads with starch, indicating the incorporation of starch into the beads’ matrices, which could have contributed to improved survival of LRGG in SGDMs [55]. Although samples with soybean oil did not show a positive impact on survivability, their spectra showed prominent C–H stretching bands at 2862–2932 cm^−1^ and a C=O stretching band at 1749 cm^−1^, consistent with the presence of lipids.

Chitosan-coated alginate beads were further studied using SIDMs for up to 4 h to evaluate the impacts of food materials on the simulated intestinal fluid at pH 8.1. Figure 5a shows the survival of LRGG still entrapped in the chitosan-coated alginate beads after 4 h of SIDMs. The results clearly indicate that the addition of food materials significantly affected the viability of LRGG during intestinal digestion. Similar to their impacts in SGDMs, the addition of casein and corn starch was found to enhance the survivability of LRGG in the simulated intestinal fluid. Casein provided the highest protection, maintaining viable LRGG at 8.50 ± 0.11 Log CFU/g beads, which is close to the initial cell count of 9.13 ± 0.12 Log CFU/g. Starch maintained the second-highest number of viable LRGG, with approximately 5.81 ± 0.44 Log CFU/g beads, representing a reduction of about 1.5 Log CFU/g from the viable LRGG count of 7.27 ± 0.34 Log CFU/g observed after 2 h of gastric digestion with corn starch. This result suggests that its protective effect may stem from physical barrier effects or increased viscosity, which reduces gastrointestinal juice diffusion into capsules [56,57]. On the other hand, no remaining cells were observed in the presence of soybean oil since all the chitosan-coated alginate beads were completely dissolved. These findings indicate that casein and starch positively influence the protection of probiotics from bile in intestinal fluid when digested simultaneously with these materials.

Figure 5b shows the accumulated number of released LRGG during 4 h of the SIDMs. The release of viable cells was most intense during the first 60 min, aligning with a previously published study on the release behavior of Bifidobacterium breve encapsulated in chitosan-coated alginate beads [58]. The addition of food materials caused a significant difference compared to the control (*p* < 0.05), except for casein (*p* > 0.05). After 4 h of digestion in SIDMs, control, starch, and casein samples reached 3.15 ± 0.03, 5.89 ± 0.70, and 3.35 ± 0.28 Log CFU/g of accumulated release cells, respectively. Although casein increased the survival of entrapped LRGG in both SGDMs and SIDMs, it did not lead to a greater release of cells during SIDMs compared to the control. In contrast, starch led to a higher release of cells, which corresponded to the number of remaining viable cells in the beads. Therefore, it is assumed that the increase in released cells is due to the enhanced survivability of LRGG observed in SIDMs. Interestingly, soybean oil resulted in no released viable cells in the intestinal juice after 4 h (<2 Log CFU/g detection limit). This might be due to the interaction of fatty acids, which are released during lipolysis in intestinal digestion and negatively charged in the intestinal juice, with calcium ions [59,60], disrupting the cross-linked alginate structure in the capsule bodies, possibly acting similarly to the activity of a chelating agent. Even though a lot of studies incorporate oils in encapsulation systems, these results demonstrate the novel finding that, unlike proteins and starches, oils exhibit completely different digestive behavior when present outside the capsules.

As shown in Figure 5c, the protective effects of casein are assumed to be due to its pH-modulating ability, which resulted in a slight acidification of the intestinal fluid to 5.78 ± 0.09 from an initial pH of 8.1 over 4 h. Starch, however, did not significantly alter the pH compared to the control (*p* > 0.05), suggesting that its protective effect is not identical to that of casein, which is primarily attributed to pH modulation. The enhancement of protectability by starch was only confirmed in alginate beads, potentially due to starch acting as a filler that occludes surface pores, thereby reducing fluid penetration and digestive enzyme access [56,57]. The effect of viscosity could also be considered for the improvement of survivability, as it was confirmed that the high viscosity hinders the diffusion of the digestive fluid [41]. Soybean oil slightly lowered the pH of intestinal juice more than the control, likely due to the hydrolysis of triacylglycerols, which produces fatty acids and glycerides [61]. Figure 5d presents the weight ratio of chitosan-coated beads after 4 h of the SIDMs. The addition of casein and soybean oil had significant effects on the morphological change in the beads compared to the control, as casein made the beads maintain a compact structure, while soybean oil completely dissolved the beads into the intestinal fluid. This observation is further supported by Figure 6, which shows that casein led to less swelling compared to the control. Given that probiotic inactivation in intestinal fluid is primarily caused by bile exposure, the addition of soybean oil resulted in the complete breakdown of chitosan-coated beads, leading to the full inactivation of LRGG due to a more challenging exposure of probiotics to bile juice.

Figure 7 shows the FTIR spectra of digested chitosan-coated beads with various food matrices in SIDMs after 4 h. Likewise, there are some peaks mutually found in the samples in Figure 4, including an O–H stretching band at 3400–3500 cm^−1^, C–H stretching bands at 2862–2885 and 2930–2940 cm^−1^, and a C=O band at 1630–1636 cm^−1^ due to the hydroxyl and carboxyl groups [51]. In addition, all spectra exhibited strong vibrational peaks between 1000 and 1600 cm^−1^, possibly due to the presence of lipase, pancreatin, and bile salts in the intestinal fluid [62]. For example, prominent peaks were observed at 1025–1036 cm^−1^ and 1401–1408 cm^−1^, which could be attributed to the S=O stretching of taurine conjugates and the COO− symmetric stretching of alginate and bile salts [63,64]. The casein-containing sample exhibited broader and more intense peaks throughout the fingerprint region (1000–1700 cm^−1^), suggesting stronger interactions between casein and intestinal enzymes compared to other samples. In contrast, samples with soybean oil did not show a peak at 1556 cm^−1^, which could be attributed to the Amide II band of digestive enzymes. This may be due to the complete dissolution of the beads in the intestinal fluid, reducing the detectability of enzyme-related features in the FTIR spectrum. Notably, the soybean oil sample was the only one observed in a fully fluid state rather than as residual bead structures.

### 4.2. Dynamic In Vitro Gastrointestinal Digestion with Continuous Juice Secretion and Emptying

To investigate the impact of dynamic in vitro GI models on entrapped LRGG during 2 h of gastric digestion, two levels of gastric pH, 2.5 or 3.0, which were commonly chosen for dynamic gastric digestion, was set for the testing, and then the observed viable number of cells, weight ratio, and pH shifts were summarized in Figure 8. The results confirmed that digestion using DGDMs significantly reduced LRGG viability in chitosan-coated alginate beads compared to digestion using SGDMs at both initial pH 2.5 (5.21 ± 0.32 vs. 6.34 ± 0.27 Log CFU/g beads) and pH 3.0 (6.61 ± 0.23 vs. 8.36 ± 0.28 Log CFU/g beads) gastric juices (*p* < 0.05). The weight ratio analysis showed that beads used in the dynamic model exhibited a significant decrease in weight at pH 2.5, from 92.0 ± 0.9% to 52.0 ± 10.0% of their initial weight (*p* < 0.05), while no significant difference was observed at pH 3.0 (*p* > 0.05). These findings indicate that the impact of dynamic gastric simulation is greater when the administered pH is lower. It is generally observed that the lower gastric pH in simulated fluid shows lower viable cells in gastric digestion [65]. For the impact of dynamic conditions, a previous study has also highlighted such differences between static and dynamic GI models in food assessments. For instance, a study comparing milk protein digestibility showed that dynamic models led to more extensive breakdown into free amino acids, likely due to continuous pepsin supply during gastric digestion [66]. In fact, the pH values after gastric digestion testing showed that DGDMs maintained pH values similar to the initial pH, likely due to the continuous supply of digestive solution (Figure 8c). In contrast, SGDMs resulted in higher pH values than the initial pH, likely due to the buffering effects of probiotic capsules. As a result, these findings suggest that evaluating probiotic capsules using static gastric models could lead to an overestimation of viability, as static models mitigate acidity during digestion testing.

Figure 9a shows the accumulated number of cells released in the simulated intestinal fluid using DIDMs, which were digested with SGDMs at pH 2.5 or pH 3.0 before intestinal digestion. As mentioned in Figure 6, the chitosan-coated beads retained their shape by swelling after 4 h of SIDMs; however, it was confirmed that the beads were completely decomposed after 60 min when digested in DIDMs. The number of released cells reached a maximum at 60 min in DIDMs at both pH 2.5 (4.55 ± 0.40 Log CFU/g beads) and pH 3.0 (7.18 ± 0.44 Log CFU/g beads). Both conditions showed about a 1–2 Log CFU/g reduction compared to the number of cells before intestinal digestion using DIDMs, which were 6.34 ± 0.27 and 8.36 ± 0.28 Log CFU/g beads with pre-treatments in the SGDMs at pH 2.5 and pH 3.0, respectively. After 60 min of DIDMs, the number of released cells remained unchanged up to 90 min (*p* > 0.05).

Figure 9b presents the pH change during DIDMs up to 90 min. The pH of samples after 90 min of DIDMs, which were pre-treated with either pH 2.5 or pH 3.0 SGDMs, reached 8.79 ± 0.09 and 8.60 ± 0.02, respectively. The pH values of DIDMs at 90 min and SIDMs at 4 h were very close when pre-treated with pH 2.5 SGDMs, suggesting more rapid degradation due to pH in DIDMs. In addition, the continuous supply of fresh digestive fluid may have reduced the concentration gradient of the previously decomposed capsules, which could have further facilitated capsule degradation during intestinal digestion.

## 5. Conclusions

This study clearly demonstrated that the selection of food matrices and the simulation of dynamic GI conditions play a critical role in the survival and release of encapsulated LRGG. Among the tested food materials, casein showed the strongest protective effect during digestion, primarily due to its pH-buffering capacity, which increased the gastric pH and maintained capsule integrity. Corn starch also contributed to improved survivability, likely through its physical barrier function or increased viscosity, although its mechanism differed from that of casein. In contrast, soybean oil led to the complete disintegration of capsules and inactivation of entrapped cells during intestinal digestion, possibly due to the interaction between fatty acids and calcium ions that disrupted the cross-linked structure. The comparison between static and dynamic GI models revealed that dynamic conditions, especially under lower pH, significantly reduced probiotic viability and promoted faster capsule degradation. These findings suggest that digestion studies using only static models may overestimate probiotic survivability by approximately 1–2 Log CFU/g. Overall, these results provide important insights into the combined effects of food matrices and GI conditions on the functionality of encapsulated probiotics. This information can contribute to the optimization of encapsulation strategies for targeted applications in functional food and nutraceutical products. Future studies should explore more complex food systems and formulations to further advance the development of effective probiotic delivery systems.

## Figures and Tables

**Figure 1 foods-14-02094-f001:**
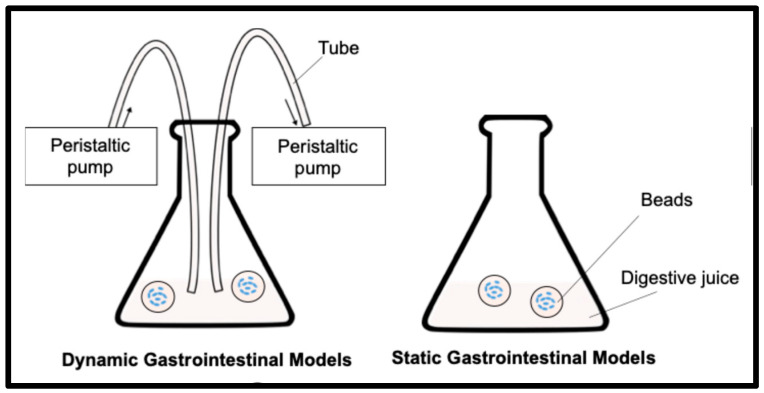
Schematic diagram of in vitro dynamic gastrointestinal models and static gastrointestinal models.

**Figure 2 foods-14-02094-f002:**
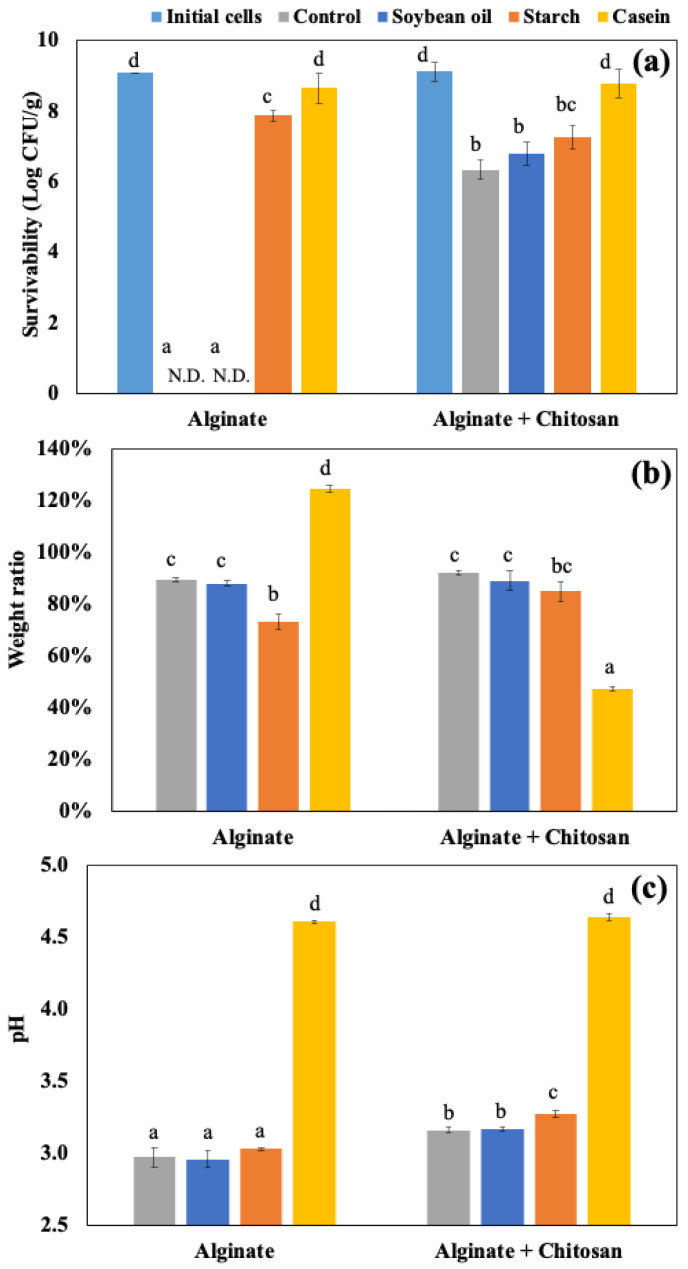
Survival of LRGG in alginate and chitosan-coated alginate beads in simulated gastric fluids at pH 2.5 for 2 h using in vitro static gastric models (SDGMs). (**a**) Survivability; (**b**) weight ratio (>100% indicates swelling and <100% indicates shrinking); (**c**) pH shifts. N.D. stands for not detected (<2 Log CFU/g detection limit). Multiple comparisons were performed within each bead type. Groups labeled with different symbols differ significantly (*p* < 0.05); identical letters indicate no significant difference.

**Figure 3 foods-14-02094-f003:**
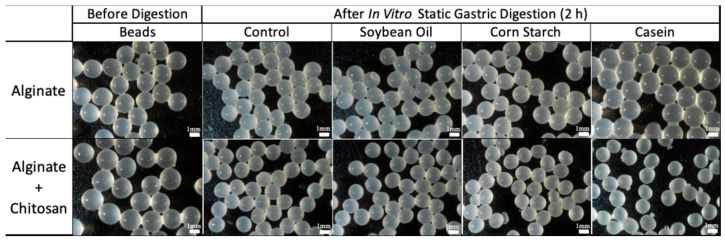
Appearance of alginate and chitosan-coated alginate beads in static gastric digestion models (SGDMs) at pH 2.5 with food matrices.

**Figure 4 foods-14-02094-f004:**
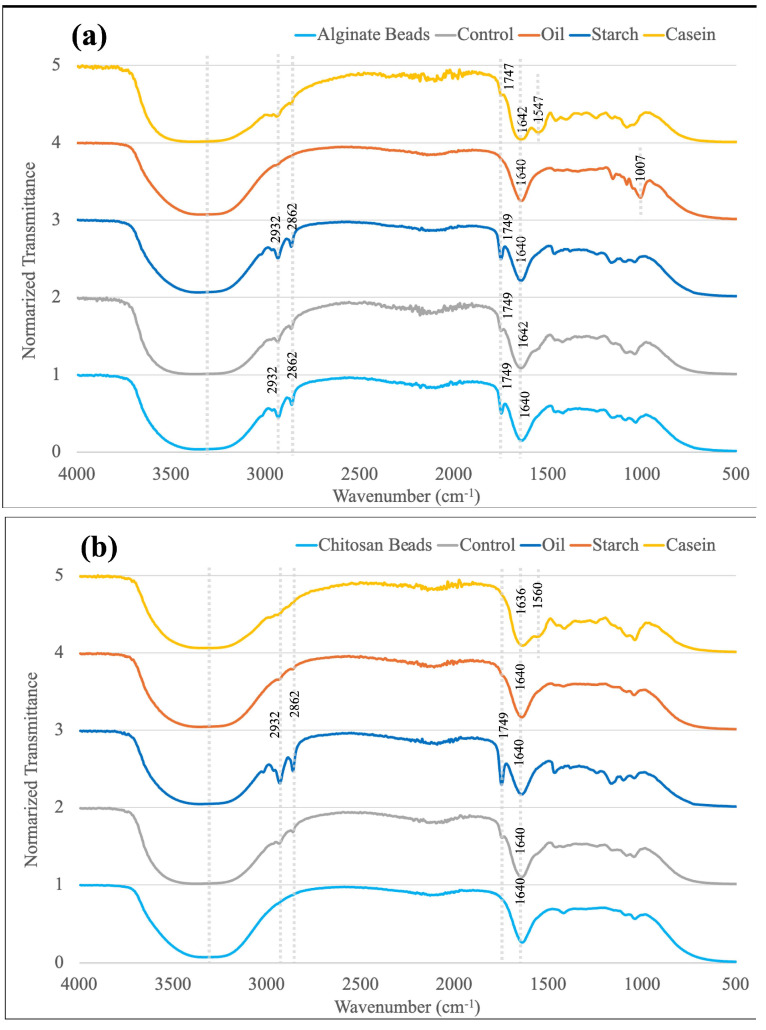
FTIR spectra of (**a**) alginate beads, (**b**) chitosan-coated alginate beads, and digested samples in static gastric digestion models (SGDMs) at pH 2.5 with food matrices.

**Figure 5 foods-14-02094-f005:**
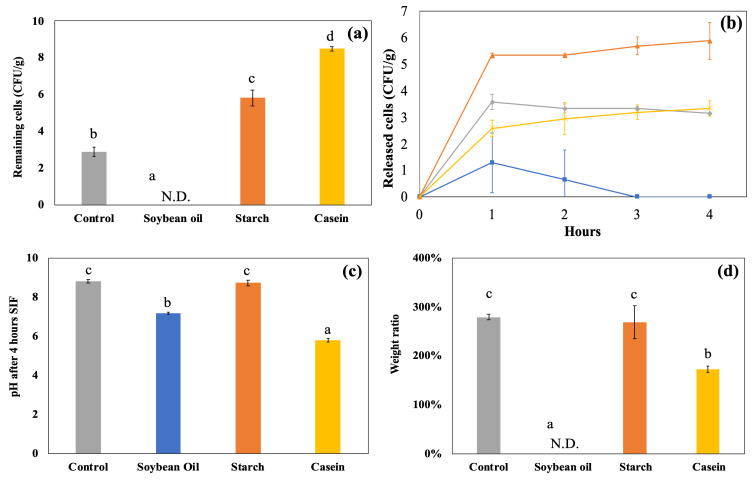
(**a**) Survival, (**b**) release, (**c**) pH shifts, and (**d**) weight ratio (>100% indicates swelling and <100% indicates shrinking) of chitosan-coated alginate beads with LRGG in simulated intestinal fluids at pH 8.1 for 4 h using in vitro static intestinal models (SIGMs). N.D. stands for not detected. Groups labeled with different symbols differ significantly (*p* < 0.05); identical letters indicate no significant difference.

**Figure 6 foods-14-02094-f006:**
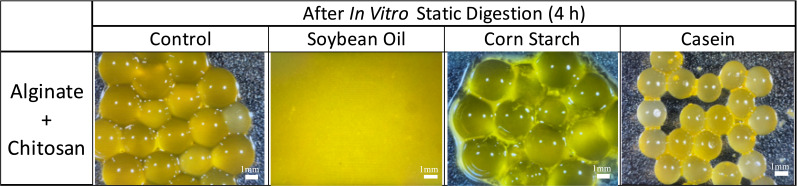
Appearance of chitosan-coated alginate beads in static intestinal digestion models (SIDMs) at pH 8.1 at 4 h with food matrices.

**Figure 7 foods-14-02094-f007:**
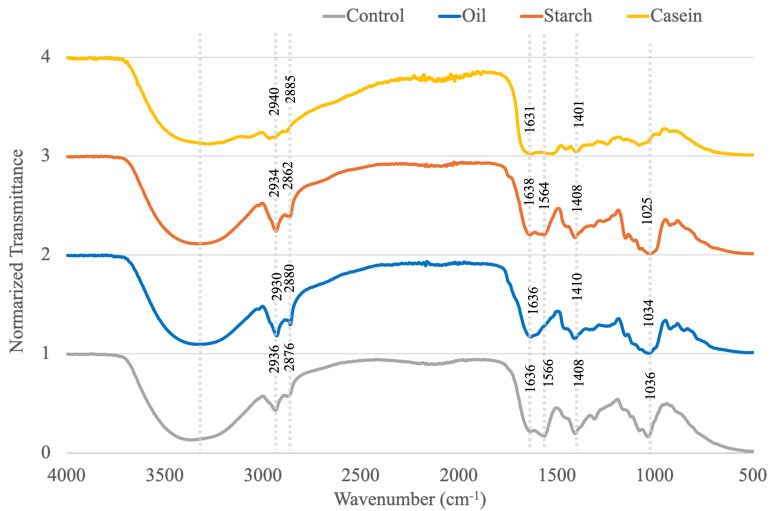
FTIR spectra of chitosan-coated alginate beads in static intestinal digestion models (SIDMs) at pH 8.1 with food matrices.

**Figure 8 foods-14-02094-f008:**
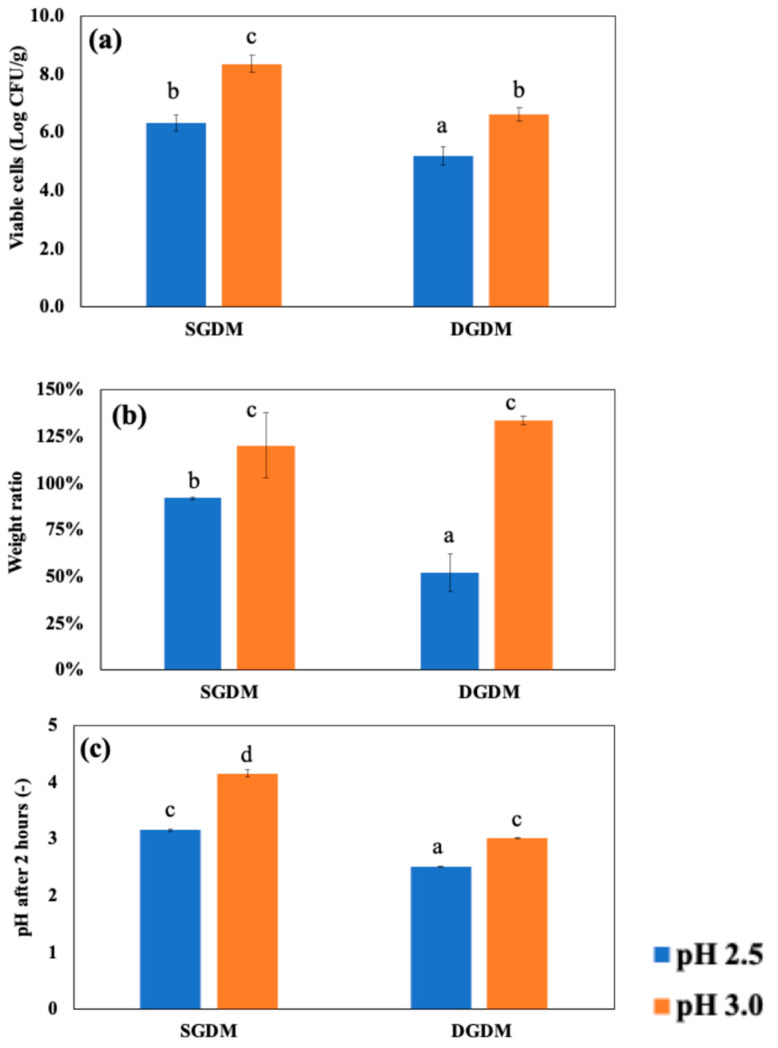
Survival of LRGG in chitosan-coated alginate beads in simulated gastric fluids at pH 2.5 or 3.0 for 2 h using in vitro static gastric models (SDGMs) and dynamic gastric digestion models (DGDMs). (**a**) Viable cells; (**b**) weight ratio (>100% indicates swelling and <100% indicates shrinking); (**c**) pH shifts. Groups labeled with different symbols differ significantly (*p* < 0.05); identical symbols indicate no significant difference.

**Figure 9 foods-14-02094-f009:**
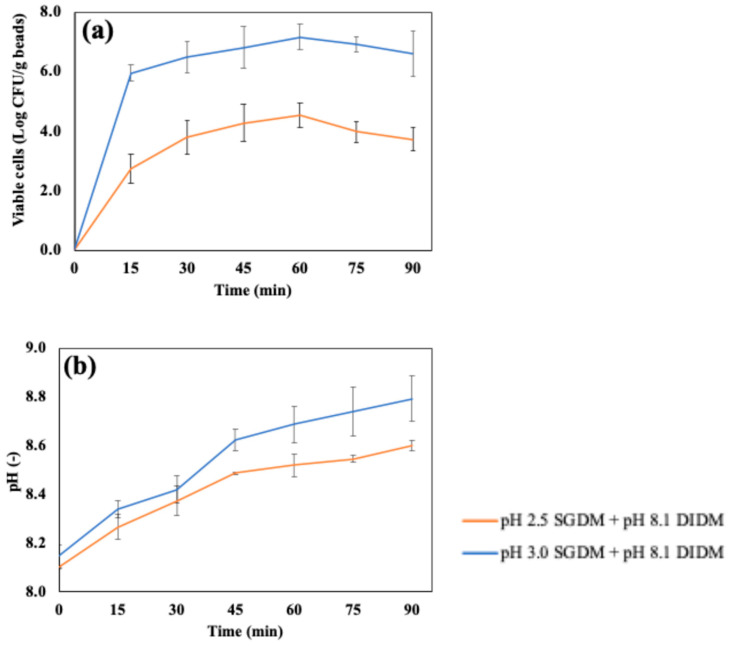
(**a**) Release of viable LRGG and (**b**) pH shifts in chitosan-coated alginate beads in simulated dynamic intestinal digestion models (DIDMs) for 4 h. Samples were treated in SGDMs for 2 h at pH 2.5 or 3.0 in advance.

## Data Availability

The original contributions presented in the study are included in the article, further inquiries can be directed to the corresponding author.

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
