# Peer review of "Survival of Lactobacillus rhamnosus GG in Chitosan-Coated Alginate Beads: Effects of Food Matrices (Casein, Corn Starch, and Soybean Oil) and Dynamic Gastrointestinal Conditionsâ€"

_foods, 2025, doi:10.3390/foods14122094_

Round 1
Reviewer 1 Report (New Reviewer)
Comments and Suggestions for Authors
The study evaluates the effects of food matrices and the impact of dynamic in vitro gastrointestinal models on the survival and release of Lacticaseibacillus rhamnosus GG during digestion simulations. Overall, the topic is relevant and the manuscript is well-organized. However, several aspects require clarification or improvement, as detailed below.
Abstract
According to the journal "Instructions for Authors", the abstract must not exceed 200 words. Please revise the abstract accordingly.
Keywords
Please provide keywords that are not already included in the title.
Methods
Lines 139–143: Please describe the methodology used to calculate the number of viable cells.
Section 3.2: Indicate whether this methodology is based on previous studies. If so, please provide the appropriate references.
Section 3.4: Similarly, clarify whether the method used is supported by previous literature.
Section 3.5: Please describe the control condition used.
Lines 231–237: Please remove this paragraph.
Results and discussion
Lines 246–256: The discussion of Figure 2a needs revision. The statement that chitosan improved survivability in all groups is not correct. According to the data, survivability appeared to increase only in the control and soybean oil groups when chitosan was added.
Figures 2a, 2b, and 2c: Clarify whether the statistical analysis was performed within each encapsulation group (alginate and alginate+chitosan) or as a comparison across all treatment means.
Line 263–264: Please provide a reference to support this statement.
Figure 3: The images labeled as “beads” are unclear. Please specify which images show beads before digestion and which are after digestion.
Lines 287–295: Please make a direct comparison with the results presented in Figure 2c. The discussion in this paragraph is unclear and needs to be better connected to the figure data.
Figure 5: The discussion is generally well-developed; however, it lacks comparison with existing literature. Please include relevant studies that have evaluated chitosan-alginate beads during digestion.
Conclusion
The conclusion is too general. Please revise this section to clearly highlight the main findings of the study.
Author Response
The study evaluates the effects of food matrices and the impact of dynamic in vitro gastrointestinal models on the survival and release of Lacticaseibacillus rhamnosus GG during digestion simulations. Overall, the topic is relevant and the manuscript is well-organized. However, several aspects require clarification or improvement, as detailed below.
- According to the journal "Instructions for Authors", the abstract must not exceed 200 words. Please revise the abstract accordingly.
Thank you for your suggestion. The abstract is edited within 200 words following the requirement.
(Line 9-) Probiotics like Lactobacillus rhamnosus GG (LRGG) offer health benefits but face reduced viability under harsh gastrointestinal (GI) conditions. Encapsulation improves stability, yet most studies rely on static GI models with simplified environment that may overestimate survival. This study assessed LRGG survival using chitosan-coated alginate beads under both static and dynamic GI models, including peristaltic flow and continuous juice replenishment. Food matrices (casein, corn starch, soybean oil) were tested in static models. Beads were prepared via extrusion and subjected to simulated gastric and intestinal digestion. After 2 and 4 hours of digestion, casein preserved LRGG viability at 8.50 ±â€¯0.11 Log CFU/g, compared to 5.81 ±â€¯0.44 with starch and undetectable levels with soybean oil. Casein’s protective effect was attributed to its pH-buffering capacity, raising gastric pH from 2.5 to 4.6. Starch offered moderate protection, while soybean oil led to bead dissolution due to destabilization of the egg-box structure. Dynamic GI models showed greater reductions in LRGG viability than static models, emphasizing the need for physiologically relevant simulations. Results highlight the importance of selecting appropriate food matrices and digestion models for accurate probiotic assessment, sup-porting improved encapsulation strategies in functional food development.
- Please provide keywords that are not already included in the title.
Thank you for your comment. We have added further keywords as follows.
Probiotics, Lactobacillus rhamnosus GG, alginate, chitosan-coating, dynamic gastrointestinal models, extrusion, cross-linking, casein, corn starch, soybean oil
- Lines 139–143: Please describe the methodology used to calculate the number of viable cells.
Thank you for your suggestion. We have performed standard plate count with serial dilution. The methodology regarding the calculation of viable cells is explained in more detail as follows.
(Line 129-) Viable LRGG cells were enumerated using a standard plate count with serial dilution, followed by incubation on MRS agar.
- Section 3.2: Indicate whether this methodology is based on previous studies. If so, please provide the appropriate references.
Thank you for your suggestion. The methodology is referred to a previous literature.
(Line 121-) Encapsulation of LRGG was performed using a previously published method with modifications [40].
- Section 3.4: Similarly, clarify whether the method used is supported by previous literature.
Thank you for your suggestion. The recipe of intestinal juice was referred to a previous literature, but the other part is originally performed.
- Section 3.5: Please describe the control condition used.
Thank you for your comment. The control condition is now explained better.
(Line 196-) Encapsulated probiotics without food matrices were also tested under the same conditions as controls.
- Lines 231–237: Please remove this paragraph.
Thank you for your suggestion. The paragraph is now removed from the manuscript.
- Lines 246–256: The discussion of Figure 2a needs revision. The statement that chitosan improved survivability in all groups is not correct. According to the data, survivability appeared to increase only in the control and soybean oil groups when chitosan was added.
Thank you very much for your comment. We have edited the explanation to discuss the data more clearly as follows.
(Line 233-) Compared to the control alginate beads, the survivability of LRGG was significantly improved with chitosan-coated capsules, consistent with previous findings [25].
- Figures 2a, 2b, and 2c: Clarify whether the statistical analysis was performed within each encapsulation group (alginate and alginate+chitosan) or as a comparison across all treatment means.
Thank you for your comment. The statistical analysis was performed within each encapsulation group, so it was further explained in the manuscript as follows.
(Line 286-) Multiple comparisons were performed within each bead type.
- Line 263–264: Please provide a reference to support this statement.
Thank you for your suggestion. Reference is now added to the statement.
(Line 257-) It is known that the isoelectric point of casein is about 4.6 [46],
- Figure 3: The images labeled as “beads” are unclear. Please specify which images show beads before digestion and which are after digestion.
Thank you for your suggestion. The images are now better explained accordingly in Figure 3 and 6.
- Lines 287–295: Please make a direct comparison with the results presented in Figure 2c. The discussion in this paragraph is unclear and needs to be better connected to the figure data.
Thank you for your suggestion. The results were further discussed and compared in the sentences accordingly.
(Line 245-) For alginate beads, casein was the only food matrix that caused the beads to swell during gastric digestion. The weight ratios relative to the original weights were 89.3 ±â€¯0.9% (control), 88.0 ±â€¯1.3% (soybean oil), 73.2 ±â€¯2.9% (starch), and 124.8 ±â€¯1.3% (casein), with casein showing a significantly higher value (p < 0.05). Notably, the addition of casein had opposite effects on alginate and chitosan-coated alginate beads, where alginate beads swelled, while chitosan-coated beads shrank after 2 hours of gastric digestion. The weight ratios of chitosan-coated beads were 92.0 ±â€¯0.9% (control), 88.9 ±â€¯3.8% (soybean oil), and 84.9 ±â€¯3.8% (starch), with no significant differences among these groups (p > 0.05). However, the addition of casein led to a significant reduction in weight ratio to 47.2 ±â€¯0.9% (p < 0.05), indicating pronounced shrinkage.
- Figure 5: The discussion is generally well-developed; however, it lacks comparison with existing literature. Please include relevant studies that have evaluated chitosan-alginate beads during digestion.
Thank you for your suggestion. We have now added a literature to compare our results.
(Line 330-) The release of viable cells was most intense during the first 60 minutes, aligning with a previously published study on the release behavior of Bifidobacterium breve encapsulated in chitosan-coated alginate beads [58].
Conclusion
- The conclusion is too general. Please revise this section to clearly highlight the main findings of the study.
Thank you for your comment. The conclusion is now explained in more detail as follows.
(Line 467-) This study clearly demonstrated that the selection of food matrices and the simulation of dynamic gastrointestinal (GI) conditions play a critical role in the survival and release of encapsulated Lactobacillus rhamnosus GG (LRGG). Among the tested food materials, casein showed the strongest protective effect during digestion, primarily due to its pH-buffering capacity, which increased the gastric pH and maintained capsule integrity. Corn starch also contributed to improved survivability, likely through its physical barrier function or increased viscosity, although its mechanism differed from that of casein. In contrast, soybean oil led to the complete disintegration of capsules and inactivation of entrapped cells during intestinal digestion, possibly due to the interaction between fatty acids and calcium ions that disrupted the cross-linked structure. The comparison between static and dynamic GI models revealed that dynamic conditions, especially under lower pH, significantly reduced probiotic viability and promoted faster capsule degradation. These findings suggest that digestion studies using only static models may overestimate probiotic survivability by approximately 1–2 Log CFU/g. Overall, these results provide important insights into the combined effects of food matrices and GI conditions on the functionality of encapsulated probiotics. This information can contribute to the optimi-zation of encapsulation strategies for targeted applications in functional food and nutraceutical products. Future studies should explore more complex food systems and formulations to further advance the development of effective probiotic delivery systems.
Reviewer 2 Report (New Reviewer)
Comments and Suggestions for Authors
The paper entitled "Survival of Lactobacillus rhamnosus GG in Chitosan-Coated Alginate Beads: Effects of Food Matrices (Casein, Corn Starch, Soybean Oil) and Dynamic Gastrointestinal Conditions" has been improved in many aspects, such as introduction, methods, results and discussion, and conclusions. The paper consists of good FTIR and diffusion studies with explanatory capsule images, which makes it very interesting and attractive for the academic, scientific and industrial reader.
Comments on the Quality of English LanguageNo comments.
Author Response
The paper entitled "Survival of Lactobacillus rhamnosus GG in Chitosan-Coated Alginate Beads: Effects of Food Matrices (Casein, Corn Starch, Soybean Oil) and Dynamic Gastrointestinal Conditions" has been improved in many aspects, such as introduction, methods, results and discussion, and conclusions. The paper consists of good FTIR and diffusion studies with explanatory capsule images, which makes it very interesting and attractive for the academic, scientific and industrial reader.
We sincerely appreciate your encouraging and thoughtful feedback. We are glad to hear that you found the revisions improved the manuscript across key sections including FTIR analysis. Thank you once again for your kind words and constructive support throughout the review process.
Reviewer 3 Report (New Reviewer)
Comments and Suggestions for Authors
The manuscript “Survival of Lactobacillus rhamnosus GG in chitosan-coated alginate beads: Effects of food matrices (casein, corn starch, soybean oil) and dynamic gastrointestinal conditions” presents an interesting approach of the viability during two types of simulated conditions of the GI of encapsulated probiotics. The study is well-designed. However, some overall recommendations are suggested as follows:
L13-15. Mention the type of encapsulation assessed.
Abstract. Is one of the encapsulated better as a probiotic supplier?
L99-100. Please, include the type of encapsulated probiotics that were evaluated in each step of the study. For example, alginate and alginate/chitosan encapsulation or just alginate coated chitosan capsules. Because the encapsulation conditions are missing.
L145-174. Mention what type of encapsulated probiotics were evaluated in each type of gastrointestinal digestion model (static and/or dynamic).
L181. Include alginate beads.
L199-201. Are 4 g or corn starch, soybean oil, or casein added to beads? Or is a model food matrix prepared with these ingredients?
L199-204. Mention what type of encapsulated probiotics were evaluated.
L231-237. Delete this information. Probably it is not part of the manuscript.
L359-361. Include a citation.
Conclusion. According to the simulated conditions of the GI evaluated in this study, which type of capsules remained the viability of probiotics and which promote the release. Is one of the encapsulated better as a probiotic supplier?
Author Response
The manuscript “Survival of Lactobacillus rhamnosus GG in chitosan-coated alginate beads: Effects of food matrices (casein, corn starch, soybean oil) and dynamic gastrointestinal conditions” presents an interesting approach of the viability during two types of simulated conditions of the GI of encapsulated probiotics. The study is well-designed. However, some overall recommendations are suggested as follows:
- L13-15. Mention the type of encapsulation assessed.
Thank you for your suggestion. It is now better explained in the sentence.
(Line 12-) This study assessed LRGG survival using chitosan-coated alginate beads under both static and dynamic GI models, including peristaltic flow and continuous juice replen-ishment. Food matrices (casein, corn starch, soybean oil) were tested in static models. Beads were prepared via extrusion and subjected to simulated gastric and intestinal digestion.
- Is one of the encapsulated better as a probiotic supplier?
Thank you for your comment. We have edited the abstract within 200 words including the short explanation regarding encapsulation.
- L99-100. Please, include the type of encapsulated probiotics that were evaluated in each step of the study. For example, alginate and alginate/chitosan encapsulation or just alginate coated chitosan capsules. Because the encapsulation conditions are missing.
Thank you for your suggestion. We have fortified the explanation regarding encapsulation conditions in more detail.
(Line 90-) One of the main aims of this study is to evaluate the effects of food matrices, such as casein, corn starch, and soybean oil, on the survivability and release of probiotics encapsulated in alginate-based capsules with chitosan-coating in GI conditions. Second, this study aims to assess chitosan-coated alginate probiotic capsules using in vitro dynamic GI models simulating continuous GI juice supply and emptying, which replicate one of the key physiological activities in the human GI tracts.
- L145-174. Mention what type of encapsulated probiotics were evaluated in each type of gastrointestinal digestion model (static and/or dynamic).
Thank you for your suggestion. We have edited the sentence to clearly indicate the used beads in each condition accordingly.
(Line 144-) 2g of chitosan-coated alginate beads were mixed with 18 mL of the simulated gastric fluid in a 50 mL Erlenmeyer flask, ~
(Line 193-) 4 g of model food materials were mixed with 2 g of alginate/chitosan-coated alginate beads and 18 mL simulated gastric fluid, and gastric digestion were performed following the described methods.
(Line 196-) After 2 hours of gastric digestion, food materials and chitosan-coated alginate beads were transferred to another Erlenmeyer flask and replenished with the simulated intestinal fluid.
- Include alginate beads.
Thank you for your comment. We only performed intestinal condition for chitosan-coated beads for both food matrices study and dynamic intestinal models since the control did not show survival cells in the previous gastric condition.
- L199-201. Are 4 g or corn starch, soybean oil, or casein added to beads? Or is a model food matrix prepared with these ingredients?
Thank you for your comment. The food matrices were added in the gastrointestinal juice, not mixed in the encapsulated materials. The methodology was further explained more clearly.
(Line 193-) 4 g of model food materials were mixed with 18 mL simulated gastric fluid, followed by adding 2 g of alginate/chitosan-coated alginate beads. Gastric digestions were performed using SGDM following the described methods above.
- L199-204. Mention what type of encapsulated probiotics were evaluated.
Thank you for your suggestion. We have now added statement to explain in more detail.
(Line 193-) 4 g of model food materials were mixed with 18 mL simulated gastric fluid, followed by adding 2 g of alginate/chitosan-coated alginate beads. Gastric digestions were performed using SGDM following the described methods above. Encapsulated probiotics without food matrices were also tested under the same conditions as controls. After 2 hours of gastric digestion, food materials and chitosan-coated alginate beads were transferred to another Erlenmeyer flask and replenished with the simulated intestinal fluid.
- L231-237. Delete this information. Probably it is not part of the manuscript.
Thank you for your suggestion. It is now removed from the manuscript.
- L359-361. Include a citation.
Thank you for your suggestion. We have added a literature in the statement.
(Line 358-) Soybean oil slightly lowered the pH of intestinal juice more than the control, likely due to the hydrolysis of triacylglycerols, which produces fatty acids and glycerides [61].
- According to the simulated conditions of the GI evaluated in this study, which type of capsules remained the viability of probiotics and which promote the release. Is one of the encapsulated better as a probiotic supplier?
Thank you for your comment. In our study, the most of experiments was conducted with chitosan-coated alginate beads, and the type of capsule is not the main focus. Instead, we have added more detailed summary about the impact of food materials and dynamic GI models in the conclusion.
(Line 467-) This study clearly demonstrated that the selection of food matrices and the simulation of dynamic GI conditions play a critical role in the survival and release of encapsulated LRGG. Among the tested food materials, casein showed the strongest protective effect during digestion, primarily due to its pH-buffering capacity, which increased the gastric pH and maintained capsule integrity. Corn starch also contributed to improved sur-vivability, likely through its physical barrier function or increased viscosity, although its mechanism differed from that of casein. In contrast, soybean oil led to the complete disintegration of capsules and inactivation of entrapped cells during intestinal digestion, possibly due to the interaction between fatty acids and calcium ions that disrupted the cross-linked structure. The comparison between static and dynamic GI models revealed that dynamic conditions, especially under lower pH, significantly reduced probiotic viability and promoted faster capsule degradation. These findings suggest that digestion studies using only static models may overestimate probiotic survivability by approxi-mately 1–2 Log CFU/g. Overall, these results provide important insights into the com-bined effects of food matrices and GI conditions on the functionality of encapsulated probiotics. This information can contribute to the optimization of encapsulation strategies for targeted applications in functional food and nutraceutical products. Future studies should explore more complex food systems and formulations to further advance the development of effective probiotic delivery systems.
Reviewer 4 Report (New Reviewer)
Comments and Suggestions for Authors
There is a paragraph at the end of the materials section that seems to be part of the template.
Author Response
There is a paragraph at the end of the materials section that seems to be part of the template.
Thank you for your comment. The sentence was removed from the manuscript.
This manuscript is a resubmission of an earlier submission. The following is a list of the peer review reports and author responses from that submission.
Round 1
Reviewer 1 Report
Comments and Suggestions for Authors
The study highlighted the challenges of probiotic survival in harsh gastrointestinal (GI) conditions with emphasis on both the role of food matrices and the differences between static and dynamic GI models.
- Abstract - how corn starch and soybean oil might interact with the encapsulation matrix which leads to dissolution.
- Abstract – Please include specific quantitative results for more concrete evidence of the effects (e.g., survival rates or Log CFU differences)
- Introduction – The flow from background to objectives can be improved. Some background details are presented in a lengthy manner that could dilute the focus on your study’s objectives. It will be clearer to highlight physiological differences captured by dynamic models (e.g., peristalsis, continuous secretion, and gradual pH changes) that are missing in static models, resulted in an overestimation of probiotic viability.
- Introduction -Authors may improve the introduction by critically reviewing available evidence on the underlying mechanisms by which different food matrices (such as casein, starch, and soybean oil) might influence the encapsulation performance i.e. how pH-buffering capacity of casein creates favorable microenvironment for probiotics. How starch acts as a physical barrier or filler that reduces the porosity of the beads. Clarify how and why soybean oil might lead to the rapid breakdown of the encapsulation matrix
- Method - The study should include the investigation on the structural changes in the beads via advanced imaging techniques. In fact, molecular interactions between the food matrices and the encapsulation material can be performed through FTIR, Raman to further elucidate the underlying reasons for the observed differences.
- Method - more detailed explanation of the setup (e.g., peristaltic pump configuration, exact timings of fluid exchanges) could enhance clarity.
- Results- can be improved by incorporating more specific numerical data (e.g., exact Log CFU differences, precise percentage changes) directly in the text. This can strengthen the reader’s understanding without relying solely on figures.
- Result – Please check all figures for comprehensive legends that explain all symbols, error bars, and statistical annotations
- Results - Provide a brief interpretation or linking the observed trends to the proposed mechanisms (such as the pH-buffering effect of casein or the rapid dissolution caused by soybean oil) within the results text.
Author Response
Comment 1:
Abstract - how corn starch and soybean oil might interact with the encapsulation matrix which leads to dissolution.
Abstract – Please include specific quantitative results for more concrete evidence of the effects (e.g., survival rates or Log CFU differences)
Response 1:
Thank you very much for your suggestion. Additional explanation and specific numbers were added in the abstract as follows:
(Line 20~) The encapsulated LRGG showed a survival rate of 8.50 ±â€¯0.11 Log CFU/g in the presence of casein after 2 hours and 4 hours of gastric and intestinal digestion, compared to a reduction to 5.81 ±â€¯0.44 Log CFU/g with starch, and no viable cells detected with soybean oil. Casein exhibited the strongest protective effect, likely due to its pH-buffering properties, raising the gastric pH from 2.5 to 4.6. Starch moderately improved viability by providing additional shielding, while soybean oil led to the complete dissolution of beads in the intestinal phase, likely due to the breakdown of the egg-box structure. Furthermore, dynamic GI models (DGDM and DIDM) showed greater reductions in LRGG viability than static GI models, highlighting the importance of using physiologically relevant digestion simulations. The findings suggest that evaluating probiotic encapsulation solely with static models may lead to the inaccuracy of survival and release rates by 1-2 Log CFU/g. This study underscores the critical role of food matrices in probiotic protection and the necessity of dynamic GI models for accurate viability assessments, contributing to the optimization of probiotic encapsulation and assessments for functional food applications.
Comment 2:
- Introduction – The flow from background to objectives can be improved. Some background details are presented in a lengthy manner that could dilute the focus on your study’s objectives. It will be clearer to highlight physiological differences captured by dynamic models (e.g., peristalsis, continuous secretion, and gradual pH changes) that are missing in static models, resulted in an overestimation of probiotic viability.
- Introduction -Authors may improve the introduction by critically reviewing available evidence on the underlying mechanisms by which different food matrices (such as casein, starch, and soybean oil) might influence the encapsulation performance i.e. how pH-buffering capacity of casein creates favorable microenvironment for probiotics. How starch acts as a physical barrier or filler that reduces the porosity of the beads. Clarify how and why soybean oil might lead to the rapid breakdown of the encapsulation matrix
Response 2:
Thank you very much for your comments. For these two comments regarding introduction, it was improved accordingly, clarifying the flow from background to objectives and explaining features of dynamic GI models and food materials more in detail. The change is highlighted in red as follows:
(Line 80~) In addition to encapsulation, the presence of food materials in the digestive environment may significantly impact probiotic survival and release. Specific food components, such as proteins, carbohydrates, and lipids may provide additional protection to probiotics by altering the local digestive environment and interacting with the encapsulation matrix. For instance, protein is known to exhibit pH-buffering capacity, potentially creating a situation that mitigates gastric acidity and then improve probiotic viability [28-30]. Starch is also a common material to improve the integrity of probiotic capsules as a filler or barrier matrix, limiting diffusion of gastrointestinal juices into the encapsulants [31-32]. Lipids are often incorporated into emulsion-based encapsulation method and freeze-dried encapsulation technology, providing a carrier for hydrophobic compounds and carbon sources for probiotic fermentation [30, 33]. However, their effects when present outside the capsules and digested simultaneously have not been thoroughly investigated. Since probiotics are typically consumed with food, simulating digestion in the presence of food materials offers a more relevant assessment of probiotic viability and supports better design of effective probiotic administration. Therefore, evaluating the role of basic food matrices in probiotic protection other than wall materials is also essential.
Moreover, static in vitro GI models have been traditionally used to evaluate the survival of encapsulated probiotics. These models provide controlled conditions, where pH, enzyme concentrations, and digestion times are maintained at predefined levels [14, 34]. In contrast, dynamic in vitro GI models offer a more realistic simulation of digestion by incorporating gradual pH changes, continuous replenishment of digestive fluids, and peristaltic movements, mimicking the actual in vivo environment more accurately [35-37]. It has been revealed that the digest ability of in vitro dynamic GI models have more relevance in that of in vivo GI models [36, 37]. There are also a few studies using dynamic GI models for assessing probiotics strains [38, 39]; however, the assessment on encapsulated probiotics was rarely conducted. While the use of dynamic GI models to evaluate food materials is gaining a huge interest in food science area, the impacts of dynamic physiological activities of GI tracts on encapsulated probiotics is yet unclear.
Hereby, this study has two main objectives to evaluate encapsulated probiotics under more advanced digestive conditions. One of the main aims of this study is to evaluate the effects of food matrices, such as casein, corn starch, and soybean oil, on the survivability and release of encapsulated probiotics in GI conditions. Second, this study aims to assess probiotic encapsulants using in vitro dynamic GI models simulating continuous GI juice supply and emptying, which replicate one of the key physiological activities in the human GI tracts. The findings from this research will provide valuable insights into the optimization of encapsulation techniques, the role of food matrices in probiotic protection, and the importance of dynamic GI models in assessing probiotic stability. This study contributes to the development of more effective probiotic formulations for functional food applications, ensuring enhanced survival and bioavailability in the human digestive tract.
Comment 3:
Method - The study should include the investigation on the structural changes in the beads via advanced imaging techniques. In fact, molecular interactions between the food matrices and the encapsulation material can be performed through FTIR, Raman to further elucidate the underlying reasons for the observed differences.
Response 3:
We truly appreciate the reviewer’s suggestion regarding the use of FTIR or Raman spectroscopy to explore structural changes and interactions. We agree that these techniques could offer valuable insights into the mechanisms underlying the observed differences. While detailed characterization is certainly important, our current study focused on evaluating the overall protective effects within a simulated digestive context, serving as a basis for future investigations. As we are continuing work on related topics, we will consider incorporating FTIR analysis in future studies to expand upon these findings. Accordingly, we have revised the discussion section to include additional literature and have also noted this point as a potential direction for further research.
Comment 4:
Method - more detailed explanation of the setup (e.g., peristaltic pump configuration, exact timings of fluid exchanges) could enhance clarity.
Response 4:
Thank you so much for your suggestion. The methodology regarding dynamic models is now explained more in detail;
(Line 207~) For the setup, a pair of peristaltic pumps were connected to sterilized tubes and the tube ends were placed in the gastric model. The other ends of the tubes were placed in gastric juice stock and sterilized conical tube for juice supply and emptied juice collection. The emptying tube was closed with a filter paper to avoid emptying the beads from the model during the test. The supplying and emptying of gastric juice were continuously performed at a rate of 2 mL/min for 2 hours.
Comment5:
Results- can be improved by incorporating more specific numerical data (e.g., exact Log CFU differences, precise percentage changes) directly in the text. This can strengthen the reader’s understanding without relying solely on figures.
Response 5:
Thank you so much for your suggestion. We acknowledged that some sentences lacked specific numbers to show the results, so the sentences were better explained accordingly.
(Line 289~) Casein increased LRGG viability from no viable cells detected to 8.65 ± 0.43 Log CFU/g in alginate beads and 6.34 ± 0.27 Log CFU/g to 8.79 ± 0.40 Log CFU/g in chitosan-coated beads (p<0.05). This was assumed to be mainly due to the significant pH modulation effects of protein component in the gastric juice, which was further discussed below. Other than casein, the viability of LRGG was significantly enhanced when the alginate beads were digested with corn starch to 7.87 ± 0.16, while cell counts dropped the detection limit (< 2 Log CFU/g) in both soybean oil and control samples.
(Line 374~) After 4 hours of digestion in SIDM, control, starch, and casein samples reached 3.15 ± 0.03, 5.89 ± 0.70, and 3.35 ± 0.28 Log CFU/g of accumulated release cells, respectively.
Comment 6:
Result – Please check all figures for comprehensive legends that explain all symbols, error bars, and statistical annotations
Response 6:
Thank you so much for your comments. The annotation of statistical analysis is added in the figure notes as follows:
Groups labeled with different symbols differ significantly (p < 0.05); identical symbols indicate no significant difference.
Response 7:
Results - Provide a brief interpretation or linking the observed trends to the proposed mechanisms (such as the pH-buffering effect of casein or the rapid dissolution caused by soybean oil) within the results text.
Response 7:
Thank you so much for your comments. The observed trends and proposed mechanisms were further explained with citation.
(Line 289~) Casein increased LRGG viability from no viable cells detected to 8.65 ± 0.43 Log CFU/g in alginate beads and 6.34 ± 0.27 Log CFU/g to 8.79 ± 0.40 Log CFU/g in chitosan-coated beads (p<0.05). This was assumed to be mainly due to its pH-buffering effect that mitigates acid stress during gastric digestion. The similar protection effects were observed when incorporated into drug delivery system with casein [42] and other proteins such as soy [43] and whey [44].
(Line 365~) Starch maintained the second-highest number of viable LRGG, with approximately 5.81 ± 0.44 Log CFU/g beads, representing a reduction of about 1.5 Log CFU/g from the viable LRGG count of 7.27 ± 0.34 Log CFU/g observed after 2 hours of gastric digestion with corn starch. This result suggests that its protective effect may stem from physical barrier effects or increased viscosity, which reduces gastrointestinal juice diffusion into capsules [49], [50].
(Line 384~) Interestingly, soybean oil resulted in no released viable cells in the intestinal juice after 4 hours (<2 Log CFU/g detection limit). This might be due to the interaction of fatty acids, which are released during lipolysis in intestinal digestion and negatively charged in the intestinal juice, with calcium ions [51], [52], disrupting the cross-linked alginate structure in the capsule bodies acting, possibly similar to the activity of chelating agent. Even though a lot of studies incorporate oils in encapsulation systems, these results demonstrate the novel finding that, unlike proteins and starches, oils exhibit completely different digestive behavior when present outside the capsules.
Reviewer 2 Report
Comments and Suggestions for Authors
Title:
Review title for clarity and relevance in field. I would make it more specific to the actual matrices tested in the study.
Abstract:
Please add more quantitative results to your paper. Also mention/specify better the methods used for testing efficacy.
Introduction:
I would mention other potential preservation matrices, for instance silver bionanocomposites and how they could be applied to probiotic preservation and encapsulation (another challenge faced in the food industry). I would recommend citing silver bionanocomposites for food packaging as an example: https://www.mdpi.com/2073-4360/15/21/4243
Better contextualise your innovation :
- What are the limitations of current technologies and what are the applications of your materials?
- What is the potential commercial applications of your technology and what is the market value for such encapsulations in $? What is the market need/demands?
- More references required in the introduction to contextualise innovation.
Methods:
Looks good !
Results & Discussion:
Overall good result and discussion. However, I think the paper lacks references supporting conclusions and variety of references contextualising the innovation vs current state of the art. We would recommend to increase to at least 50 citations of original recent works (within the past 5 years) in the field in order to improve the critical discussions in the paper.
We would also suggest if possible further characterisation of the polymer beads – SEM to asses porosity of beads could also impact dissolution rates within the bead matrices.
Conclusion:
What are the applications of your novel material? For which industries? What are the planned future works and next steps for your research?
Add Nomenclature at the end of the paper with each abbreviation used in the manuscript.
Please answer these questions and queries above and resubmit for review. The most critical aspect is the low amount of references supporting the work.
Author Response
Comment 1:
Review title for clarity and relevance in field. I would make it more specific to the actual matrices tested in the study.
Response 1:
Thank you for your suggestion. The title is slightly modified to include the actual food matrices.
Survival of Lactobacillus rhamnosus GG in Chitosan-Coated Alginate Beads: Effects of Food Matrices (Casein, Corn Starch, Soybean Oil) and Dynamic Gastrointestinal Conditions
Comment 2:
Please add more quantitative results to your paper. Also mention/specify better the methods used for testing efficacy.
Response 2:
Thank you so much for you suggestions. The quantitative results and specified methodology were added in the abstract as follows:
Probiotics, such as Lactobacillus rhamnosus GG (LRGG), provide numerous health benefits, but their viability is challenged by harsh gastrointestinal (GI) conditions. Encapsulation is a promising approach to enhance probiotic stability; however, most studies rely on static in vitro GI models with simplified environments, which may overestimate probiotic survival due to the absence of dynamic physiological factors. This study evaluates the effects of food matrices and the impacts of dynamic in vitro GI models on LRGG survival and release during digestion simulations. Chitosan-coated alginate beads were prepared via extrusion method and subjected to simulated gastric and intestinal digestion using both static and dynamic models, including peristaltic flow and continuous juice replenishment at 2 ml/min flow rate. Model food matrices (casein, corn starch, and soybean oil) were also added in the static digestion models. The encapsulated LRGG showed a survival rate of 8.50 ±â€¯0.11 Log CFU/g in the presence of casein after 2 hours and 4 hours of gastric and intestinal digestion, compared to a reduction to 5.81 ±â€¯0.44 Log CFU/g with starch, and no viable cells detected with soybean oil. Casein exhibited the strongest protective effect, likely due to its pH-buffering properties, raising the gastric pH from 2.5 to 4.6. Starch moderately improved viability by providing additional shielding, while soybean oil led to the complete dissolution of beads in the intestinal phase, likely due to the breakdown of the egg-box structure. Furthermore, dynamic GI models (DGDM and DIDM) showed greater reductions in LRGG viability than static GI models, highlighting the importance of using physiologically relevant digestion simulations. The findings suggest that evaluating probiotic encapsulation solely with static models may lead to the inaccuracy of survival and release rates by 1-2 Log CFU/g. This study underscores the critical role of food matrices in probiotic protection and the necessity of dynamic GI models for accurate viability assessments, contributing to the optimization of probiotic encapsulation and assessments for functional food applications.
Comment 3:
I would mention other potential preservation matrices, for instance silver bionanocomposites and how they could be applied to probiotic preservation and encapsulation (another challenge faced in the food industry). I would recommend citing silver bionanocomposites for food packaging as an example: https://www.mdpi.com/2073-4360/15/21/4243
Response 3:
Thank you for your recommendation. The literature was added by highlighting the potential application of nanomaterials for encapsulation system in the future in the introduction part as follows:
(Line 61~) A trend of encapsulation with nanomaterials is another emerging area of study, particularly for enhancing antimicrobial and functional properties of packaging and delivery systems. For instance, bionanocomposites such as silver nanoparticles have been investigated for active food packaging applications due to their shelf-life extending and protective functions, which may inform future directions in probiotic delivery as well [21].
Comment 4:
Better contextualise your innovation :
- What are the limitations of current technologies and what are the applications of your materials?
- What is the potential commercial applications of your technology and what is the market value for such encapsulations in $? What is the market need/demands?
- More references required in the introduction to contextualise innovation.
Response 4:
Thank you so much for your comments and suggestions. In the introduction, sentences regarding market trends, limitations on dynamic GI models and food matrices are elaborated. References are added accordingly.
Encapsulation involves enclosing probiotics within protective materials to shield them from adverse external factors, such as acidity, oxidation, and mechanical stress [14-16]. From a commercial perspective, encapsulated probiotics have vast potential across functional foods, nutraceuticals, and pharmaceuticals. The global probiotic market was valued at over USD 87.7 billion in 2023 and is projected to exceed USD 100 billion by 2030, with encapsulated probiotics comprising a growing share due to demand for targeted delivery systems and shelf-stable products[17, 18]. There is a significant market demand for encapsulation solutions that extend probiotic viability during processing, storage, and digestion, especially in non-dairy, plant-based, or shelf-stable formulations. Various encapsulation techniques have been explored, including spray drying, extrusion, emulsion-based methods, and hydrogel entrapment, each offering different levels of protection and controlled release [19, 20]. A trend of encapsulation with nanomaterials is another emerging area of study, particularly for enhancing antimicrobial and functional properties of packaging and delivery systems. For instance, bionanocomposites such as silver nanoparticles have been investigated for active food packaging applications due to their shelf-life extending and protective functions, which may inform future directions in probiotic delivery as well [21]. Alginate-based encapsulation is among the most widely studied approaches due to its biocompatibility, cost-effectiveness, and ability to form gel matrices that enhance probiotic stability with versatile, extensive approaches [22-24]. Furthermore, chitosan-coated alginate beads provide additional protection, as chitosan enhances resistance to acidic environments and bile salts, improving probiotic survival during digestion [22, 25-27].
In addition to encapsulation, the presence of food materials in the digestive environment may significantly impact probiotic survival and release. Specific food components, such as proteins, carbohydrates, and lipids may provide additional protection to probiotics by altering the local digestive environment and interacting with the encapsulation matrix. For instance, protein is known to exhibit pH-buffering capacity, potentially creating a situation that mitigates gastric acidity and then improve probiotic viability [28-30]. Starch is also a common material to improve the integrity of probiotic capsules as a filler or barrier matrix, limiting diffusion of gastrointestinal juices into the encapsulants [31], [32]. Lipids are often incorporated into emulsion-based encapsulation method and freeze-dried encapsulation technology, providing a carrier for hydrophobic compounds and carbon sources for probiotic fermentation [30, 33]. However, their effects when present outside the capsules and digested simultaneously have not been thoroughly investigated. Since probiotics are typically consumed with food, simulating digestion in the presence of food materials offers a more relevant assessment of probiotic viability and supports better design of effective probiotic administration. Therefore, evaluating the role of basic food matrices in probiotic protection other than wall materials is also essential.
Moreover, static in vitro GI models have been traditionally used to evaluate the survival of encapsulated probiotics. These models provide controlled conditions, where pH, enzyme concentrations, and digestion times are maintained at predefined levels [14, 34]. In contrast, dynamic in vitro GI models offer a more realistic simulation of digestion by incorporating gradual pH changes, continuous replenishment of digestive fluids, and peristaltic movements, mimicking the actual in vivo environment more accurately [35-37]. It has been revealed that the digest ability of in vitro dynamic GI models have more relevance in that of in vivo GI models [36, 37]. There are also a few studies using dynamic GI models for assessing probiotics strains [38, 39]; however, the assessment on encapsulated probiotics was rarely conducted. While the use of dynamic GI models to evaluate food materials is gaining a huge interest in food science area, the impacts of dynamic physiological activities of GI tracts on encapsulated probiotics is yet unclear.
Hereby, this study has two main objectives to evaluate encapsulated probiotics under more advanced digestive conditions. One of the main aims of this study is to evaluate the effects of food matrices, such as casein, corn starch, and soybean oil, on the survivability and release of encapsulated probiotics in GI conditions. Second, this study aims to assess probiotic encapsulants using in vitro dynamic GI models simulating continuous GI juice supply and emptying, which replicate one of the key physiological activities in the human GI tracts. The findings from this research will provide valuable insights into the optimization of encapsulation techniques, the role of food matrices in probiotic protection, and the importance of dynamic GI models in assessing probiotic stability. This study contributes to the development of more effective probiotic formulations for functional food applications, ensuring enhanced survival and bioavailability in the human digestive tract.
Comment 5:
Overall good result and discussion. However, I think the paper lacks references supporting conclusions and variety of references contextualising the innovation vs current state of the art. We would recommend to increase to at least 50 citations of original recent works (within the past 5 years) in the field in order to improve the critical discussions in the paper.
We would also suggest if possible further characterisation of the polymer beads – SEM to asses porosity of beads could also impact dissolution rates within the bead matrices.
Response 5:
Thank you so much for your suggestion. The manuscript is now supported with 50+ literatures for contextualization and better discussion.
We also appreciate the reviewer’s suggestion regarding the further characterization of the polymer beads using SEM to assess surface morphology and porosity. We fully acknowledge that SEM may play a significant role in observing morphological attributes of beads during testing. While SEM analysis was not within the current study, we agree that incorporating such microstructural characterization would provide deeper insights and will strongly consider including this in our future investigations.
Comment 6:
What are the applications of your novel material? For which industries? What are the planned future works and next steps for your research?
Thank you so much for your comments. The conclusion is improved highlighting the future application and next steps for our research as follows:
(Line 507~) This study highlights the critical role of food matrices and simulation of dynamic gastrointestinal digestion for a more in-depth assessment of probiotics in alginate-based capsules. The addition of casein protected the entrapped probiotics during gastrointestinal digestion due to their pH-buffering properties. Starch also exhibited slight protective effects, although its mechanism is expected to be distinct from that of casein. On the other hand, soybean oil suggested the acceleration of capsule breakdown and cell inactivation during intestinal digestion. As these basic food ingredients are commonly included as a building block of food products, these findings suggest that incorporating appropriate food components alongside encapsulated probiotics can further improve or deteriorate their viability and potential health benefits. The comparison between static and dynamic gastrointestinal models indicated more severe damage and release will occur in probiotic capsules, revealing the importance of continuous digestive fluid circulation during digestion studies. These insights contribute to the design of more effective probiotic formulations in the functional food, nutraceutical, and pharmaceutical industries, where targeted delivery and viability are crucial for health claims and consumer efficacy. Future research should explore the effects of more complex food matrices and products, assessment using more realistic dynamic GI simulators, and thereby the advancement of encapsulation systems to optimize the processing and formulation based on the desired food application. In addition, it is also expected to incorporate further physicochemical characterization techniques, such as FTIR and SEM, to deepen the understanding of capsule interactions with digestive environments.
Comment 7:
Add Nomenclature at the end of the paper with each abbreviation used in the manuscript.
Response 7:
Abbreviations are summarized at the end of the manuscript.
Round 2
Reviewer 2 Report
Comments and Suggestions for Authors
I am really happy with the updated manuscript - happy to proceed with publication. Well done !